# COVID-19 Accelerated Cognitive Decline in Elderly Patients with Pre-Existing Dementia Followed up in an Outpatient Memory Care Facility

**DOI:** 10.3390/jcm12051845

**Published:** 2023-02-25

**Authors:** Lucia Merla, Maria Cristina Montesi, Jessica Ticali, Bruno Bais, Alessandro Cavarape, GianLuca Colussi

**Affiliations:** 1Geriatric Medicine Residency Program, Division of Internal Medicine, Department of Medicine, University of Udine, 33100 Udine, Italy; 2Department of Continuity of Care, Geriatric and Memory Care Facility, Azienda Sanitaria Universitaria Giuliano Isontina, 34132 Trieste, Italy; 3Division of Internal Medicine, “Santa Maria della Misericordia” Academic Hospital, Azienda Sanitaria Universitaria Friuli Centrale, 33100 Udine, Italy; 4Division of Internal Medicine, “Santa Maria degli Angeli” City Hospital, Azienda Sanitaria Friuli Occidentale, 33170 Pordenone, Italy

**Keywords:** SARS-CoV-2, mental status, dementia tests, activities of daily living, propensity score, observational study

## Abstract

Introduction: Coronavirus disease 2019 (COVID-19) may affect the cognitive function and activities of daily living (ADL) of elderly patients. This study aimed to establish the COVID-19 effect on cognitive decline and the velocity of cognitive function and ADL changes in elderly patients with dementia followed up in an outpatient memory care facility. Methods: In total, 111 consecutive patients (age 82 ± 5 years, 32% males) with a baseline visit before infection were divided into those who had or did not have COVID-19. Cognitive decline was defined as a five-point loss of Mini-Mental State Examination (MMSE) score and ADL comprising basic and instrumental ADL indexes (BADL and IADL, respectively). COVID-19 effect on cognitive decline was weighted for confounding variables by the propensity score, whereas the effect on change in the MMSE score and ADL indexes was analyzed using multivariate mixed-effect linear regression. Results: COVID-19 occurred in 31 patients and a cognitive decline in 44. Cognitive decline was about three and a half times more frequent in patients who had COVID-19 (weighted hazard ratio 3.56, 95% confidence interval 1.50–8.59, *p* = 0.004). The MMSE score lowered on average by 1.7 points/year, independently of COVID-19, but it lowered twice faster in those who had COVID-19 (3.3 vs. 1.7 points/year, respectively, *p* < 0.050). BADL and IADL indexes lowered on average less than 1 point/year, independently of COVID-19 occurrence. Patients who had COVID-19 had a higher incidence of new institutionalization than those who did not have the disease (45% versus 20%, *p* = 0.016, respectively). Conclusions: COVID-19 had a significant impact on cognitive decline and accelerated MMSE reduction in elderly patients with dementia.

## 1. Introduction

Several studies suggested that in elderly patients with cognitive impairment, coronavirus disease 2019 (COVID-19) is associated with cognitive function decline and worsened activity of daily living (ADL) [1,2]. However, these studies did not clearly define the extent and timing of cognitive decline and its consequences, such as institutionalization in patients with dementia. Although the reasons for this association could be multifactorial [3], dissecting the COVID-19 impact on cognitive functions beyond government-imposed social restriction could improve our strategies to prevent cognitive impairment, loss of ADL, and institutionalization in these patients.

The social isolation imposed to prevent the spread of severe acute respiratory syndrome coronavirus 2 (SARS-CoV-2) infection has deprived patients with cognitive impairment of the care and support necessary to ensure adequate mental, physical, and social well-being, conditions that are essential to maintain cognitive functions and preserve normal ADL in elderly patients with cognitive impairment [4,5]. It has been observed that social isolation and physical inactivity for over 6 months were associated with impaired cognitive functions and neuropsychiatric symptoms in more than half of patients with dementia [6]. In addition, loneliness during the lockdown led to deterioration in cognitive performance, loss of independence, and increased frailty levels, contributing to the impairment of ADL in patients with dementia [5,7,8]. 

Evidence supports both the direct and indirect effects of COVID-19 on cognitive function. COVID-19 was associated with risk factors that can accelerate the decline in cognitive function and ADL such as hospitalization, prolonged bed rest, and use of medical therapies [9,10]. In addition, electroencephalographic alterations have been documented in patients with COVID-19 and such alterations have been associated with long-term cognitive impairment [11]. Additionally, patients with COVID-19 had a greater incidence of cerebral vascular and inflammatory events with adverse functional long-term sequelae on cognitive function [12,13]. 

Assessing the preliminary effect of COVID-19 on cognitive function and indexes of ADL over time is also essential to design appropriate-sized prospective studies on the short- and long-term consequences of COVID-19 in patients with pre-existing dementia. For this purpose, this study aimed to explore the COVID-19 effect on cognitive function, ADL indexes, and new institutionalization in patients with dementia followed up in an outpatient memory care facility in Trieste city in northeastern Italy. In this location, the pandemic burden on older people was dramatic, and sensitivity to cognitive decline prevention was high [14,15]. Patients who had COVID-19 were compared with a control group of patients who were followed up during the same period but were not infected by SARS-CoV-2. Both groups of patients were subjected to the same social restrictions because of the pandemic.

## 2. Methods

### 2.1. Patients

Consecutive outpatients who were evaluated at the Memory Care Facility of the Department of Continuity of Care in Trieste were included in this retrospective longitudinal study. The inclusion criteria were all sexes, age 65 years or higher, any diagnosis of dementia, and at least a baseline and a consecutive follow-up visit. All the follow-up visits were performed during the COVID-19 pandemic, whereas the baseline visits were performed before or during the pandemic period. Specifically, we included patients whose first follow-up visit was during the pandemic and collected information about the precedent baseline visit whenever it was performed. Usually, patients received a 3- or 6-month follow-up to check therapy control or changes in clinical symptoms depending on the severity of dementia. A yearly follow-up was prescribed for clinically stable patients. When a patient missed a follow-up visit, she or he received advice to reserve a new control. Visits at the facility were standardized and conducted by an expert geriatrician who completed clinical and multidimensional evaluations. The exclusion criteria were a baseline Mini-Mental State Examination (MMSE) score of lower than 5 points or a lack of complete data. In this cohort of patients, the cases were those patients who were infected with SARS-CoV-2 and developed COVID-19 after the baseline visit, whereas the controls were those patients who did show signs of SARS-CoV-2 infection during the same period and were not tested or tested negative. The occurrence of COVID-19 and hospitalization for any cause during the study period were assessed by directly interviewing the patient or its caregiver, checking medical records, or contacting the general practitioner. All the patients who had been examined in the facility during the pandemic had to test negative before entry. Hospitalization for any cause was defined as any access to the hospital through the emergency department because of an acute illness, including severe COVID-19. Paucisymptomatic patients were not hospitalized. Patients without symptoms of SARS-CoV-2 infection were not tested unless they were contacted with COVID-19 cases. The diagnosis of SARS-CoV-2 infection relied on the positivity of two rapid antibody or genomic molecular tests, according to the World Health Organization guidelines [16]. In each patient, several factors were evaluated, namely baseline demographic data; housing characteristics; and general clinical variables including cardiovascular risk factors, comorbidities, and the number and type of common central nervous system drugs taken (i.e., use of memantine or anticholinergics, antipsychotics, antidepressants, or benzodiazepines). The presence of cerebrovascular disease was assessed by checking the history of either transitory ischemic attack or stroke and that of cardiovascular disease by either coronary/peripheral artery disease, chronic heart failure, or atrial fibrillation. Chronic kidney disease was defined by an estimated glomerular filtration rate adjusted for the body surface area lower than 60 mL/min/1.73 m^2^ at initial assessment and calculated with the Chronic Kidney Disease Epidemiology Collaboration (CKD-EPI) equation [17]. Parkinson’s disease was diagnosed according to the clinical diagnostic criteria of the Movement Disorder Society [18] and confirmed by an expert neurologist or when the patient was taking anti-Parkinson drugs. For classification, the patient’s housing was divided into home living with or without minimal caregiver help (self-sufficient patient), home living with nursing assistance, and living in a nursing home or residential facility (institutionalized patient). Patients in the last two housing conditions were considered as not-self-sufficient. Changes in cognitive functions were evaluated by calculating the MMSE score, whereas changes in daily performance were assessed by calculating the indexes of basic and instrumental ADL (BADL and IADL, respectively) at baseline and follow-up visits. The occurrence of new institutionalization after the baseline visit was also considered a marker of the patient’s general worsening and the need for major assistance. Information about demographic, housing, clinical, and central nervous system drugs was collected from electronic records or through direct interviews with patients, their relatives, caregivers, or general practitioners. The endpoints of this study were to assess whether COVID-19 was associated with a different time to cognitive decline and whether it had a different effect on the yearly change in the MMSE score and ADL indexes, as well as its effect on the new institutionalization of patients. We defined a significant cognitive decline as an MMSE score reduction of at least 5 points according to Doodle et al. [19].

Data were derived from the routine clinical management of patients performed at the memory care facility according to good clinical practice and under the Declaration of Helsinki principles. A generic informed consent form with the possibility of analyzing personal data after anonymization for research was signed by each patient or their caregiver when the patient could not give informed consent at the first contact with the center. The Institutional Review Board of the University of Udine approved this study (protocol number 153/2022), stating that no additional informed consent was needed for the retrospective analysis of data. 

### 2.2. Dementia Diagnosis, Cognitive Function, and ADL Assessment of Patients

The diagnosis of dementia was established by the criteria proposed in the Diagnosis and Statistical Manual of Mental Disorder 5th edition (DSM-V), and dementia was differentiated into the following forms: Alzheimer’s disease, vascular dementia, mixed, dementia secondary to Parkinson’s disease, dementia with Lewy bodies, and frontotemporal dementia [20,21]. Because of the low prevalence, the last three diagnoses were included in the “other” category. The MMSE was calculated according to the original Folstein et al. 30-point scale [22], adjusted for age and educational state [23]. The items assessed in the MMSE included orientation (10 points), memory (6 points), attention/concentration (5 points), language (8 points), and visuospatial function (1 point). The BADL index was a 6-point scale calculated according to Katz et al. by attributing the highest functional levels (either 0 or 1) to each of the following activities: bathing, dressing, toileting, transferring, continence, and feeding [24]. The IADL index was an 8-point scale calculated according to Lawton and Brody by attributing the highest functional level (either 0 or 1) to each of the following activities: telephone use, shopping, food preparation, housekeeping, mode of transportation, responsibility for own medications, and ability to handle finances [25]. 

### 2.3. Statistical Methods

Continuous variables were summarized as mean ± standard deviation when the variable was normally distributed or as median (interquartile range IQR) when it was not. Normal distribution was assessed by looking at the histogram of variable distribution and confirmed with the Shapiro–Wilk test. Categorical variables were summarized as counts and percentages. The variables were grouped according to the occurrence of COVID-19 or a significant cognitive decline (reduction in the MMSE score of at least 5 points). The difference between means was assessed using Student’s *t*-test for the variables normally distributed. Wilcoxon’s test was used to assess differences in the continuous variables that were not normally distributed. The difference between proportions was assessed using Fisher’s exact test. 

The effect of COVID-19 on the time to cognitive decline was assessed using the Cox proportional hazard regression and expressed as a hazard ratio (HR) with a 95% confidence interval (CI). To account for unbalanced confounders, the HR was corrected and reported as weighted HR (wHR) with a 95% CI. Weights corrected the confounding variables that predicted either COVID-19 or significant cognitive decline by a probability (*p*) lower than 10% at the univariate Cox regression analysis (including age, sex, and baseline MMSE score) and were estimated using the nonparametric covariate balancing propensity score method according to Fong et al. [26]. The probabilities of developing a significant cognitive decline in patients who had or did not have COVID-19 were represented by the Kaplan–Meier curves and compared using the nonparametric log-rank test. The cumulative incidence of COVID-19 or cognitive decline in patients who had or did not have COVID-19 was represented by cumulative hazard curves. 

The effect of COVID-19 on the yearly change in the MMSE score and ADL indexes was assessed through linear mixed-effect regression including the interaction between COVID-19 and follow-up time. The effects of COVID-19 on the MMSE score and ADL indexes were adjusted for those variables that differed between patients who had or did not have COVID-19, with a *p* lower than 10%, including age and sex. A *p* lower than 5% was considered statistically significant for rejecting the null hypothesis in statistical tests. Statistical analysis was performed with the free software R (version 4.1.3, R Core Team, R Foundation for Statistical Computing, Vienna, Austria).

## 3. Results

In this study, 111 consecutive patients were included after having excluded 5 patients because of a baseline MMSE score lower than 5 points (3 patients) or incomplete data (2 patients). The baseline characteristics of the cohort are summarized in Table 1. All the patients were older than 65 years, about two-thirds were females, and the four most frequent comorbidities were hypertension, dyslipidemia, cardiovascular disease, and diabetes. About one-fourth of the patients were self-sufficient, and about one in every sixteen was institutionalized. Antidepressants, antipsychotics, and benzodiazepines were, in that order, the three most used central nervous system drugs. Mixed was the most frequent diagnosis of dementia. After the baseline visit, a significant cognitive decline occurred in 44 patients, and of those, 19 patients had COVID-19 and 25 did not. COVID-19 occurred in 31 patients after the baseline visit, and 6 patients were hospitalized in this group. No hospitalization was observed in patients who did not have COVID-19. All the patients with COVID-19 in this study were at their first documented experience of the disease. Patients who had COVID-19 had a baseline visit between October 2018 and April 2022, and their follow-up visits were between July 2020 and July 2022, whereas those who did not have COVID-19 had the baseline visit in the facility between October 2016 and December 2021, and follow-up between June 2020 and May 2022. The estimated median time at which 50% of the cohort developed a significant cognitive decline was 2.4 years (95% CI 1.5–3.6). In those patients who had COVID-19, this median time was 1.1 years (95% CI 0.8–1.5), whereas in those who did not have COVID-19, it was 3.5 years (95% CI 2.4–4.2) (Figure 1). The overall median follow-up time of the study was 1.1 years (IQR 0.7–1.7). The median time from COVID-19 diagnosis to the follow-up visit was 3.7 months (IQR 2.0–5.4).

### 3.1. Predictors of COVID-19 and Cognitive Decline

At the baseline, the patients who had COVID-19 had a higher proportion of diabetes and institutionalization, lower MMSE scores, BADL and IADL indexes, and shorter follow-up times than controls (Table 1). The patients who developed cognitive decline were less self-sufficient, had COVID-19, and were more often hospitalized than the patients who did not have a cognitive decline (Table 2). The baseline variables that predicted COVID-19 occurrence were diabetes and institutionalization, lower BADL or IADL indexes, and lower MMSE scores (Table 3). The predictors of significant cognitive decline were the total number of drugs used, memantine use, COVID-19, hospitalization, and lower BADL and IADL indexes (Table 3). The results of balancing confounding variables that predicted either COVID-19 or cognitive decline, using propensity score weights, are reported in Figure 2. The variables weighted in the propensity score were age, male sex, hypertension, diabetes, baseline and new institutionalization, BADL and IADL indexes, the number of drugs, history of cerebrovascular disease, self-sufficiency, memantine use, and baseline MMSE. After applying the weights, having had COVID-19 was associated with a higher rate of cognitive decline over time than not having had the disease (wHR 3.56, 95% CI 1.50–8.49, *p* = 0.004). Figure 3 reports the individual cumulative hazards and 95% CIs of developing COVID-19 after the baseline visit and developing a cognitive decline in patients who had or did not have COVID-19. A higher rate of cognitive decline was observed in patients who had COVID-19.

### 3.2. Effect of COVID-19 on the Yearly Change in MMSE Score and ADL Indexes

The linear mixed-effect regression analysis unadjusted for confounders showed that the MMSE score lowered on average by 1.7 points each year, independently of COVID-19. There was an interaction between COVID-19 and follow-up time in that the patients who had COVID-19 had 1.6 points relatively greater reduction in their MMSE score each year than those patients who did not have the disease (Table 4A). The MMSE scores of patients who had COVID-19 were reduced by 3.3 points/year, and the scores of those who did not have the disease were reduced by 1.7 points/year (Figure 4). In the cohort, independently of COVID-19, there was on average a yearly reduction in BADL and IADL of 0.5 and 0.8 points, respectively. The interaction between COVID-19 and follow-up time was not relevant for BADL and IADL indexes (Table 4A, Figure 4). The confounding variables included in the model were baseline age, sex, diabetes, baseline institutionalization, new institutionalization, and all-cause hospitalization. Linear mixed-effect models, not including hospitalization, were independent of the confounding variables (Table 4B). After including hospitalization in the model, only the interaction between the COVID-19 effect and follow-up time persisted for the MMSE score (Table 4C). The follow-up time was a strong independent predictor of the reduction in the MMSE score and ADL indexes after all adjustments. Figure 4 reports the yearly changes in the MMSE score and ADL indexes for each patient who had or did not have COVID-19. The negative slope of the regression line for the relationship between the MMSE score and follow-up time was steeper in patients who had COVID-19, whereas the negative slopes of BADL and IADL indexes overlapped between the groups (Figure 4).

## 4. Discussion

We defined a significant cognitive decline as the reduction of at least five points in the MMSE score, according to Doody et al. [19]. An MMSE score reduction to such an extent in patients with dementia was considered clinically meaningful by authors because it considered the high score variability during a typical year and represented about 2 SD of the average monthly variation previously observed in patients with Alzheimer’s disease [27]. Therefore, the occurrence of cognitive decline in our study meant a clinically relevant reduction in cognitive function in patients with pre-existing dementia. 

In this study, having had COVID-19 predicted a faster cognitive decline than not having had the infection. Our results are in line with the previous findings summarized in the meta-analysis of Crivelli et al. [1]. This meta-analysis of over 2000 patients observed a decline in cognitive function in COVID-19 patients from the acute phase of the disease to 6 months after recovery compared with a control group without infection. Meta-regression analysis showed that increased age correlated with a larger COVID-19 effect [1]. In the cross-sectional study of Liu et al. on older patients and their uninfected spouses as controls, COVID-19 was associated with a higher proportion of patients with current cognitive impairment and with a longitudinal cognitive decline, at six months after recovery [28]. Our study confirms such previous results, and in addition, it preliminary defines the extent and timing of cognitive decline and its consequences, such as institutionalization, in outpatients with dementia, beyond the effect of government-imposed social restriction.

In a larger prospective study by Liu et al. on over 3000 older patients and their uninfected spouses as controls, COVID-19 was associated with a progressive cognitive decline at 6 and 12 months after recovery [29]. The authors observed that cognitive decline depends on COVID-19 severity because severe COVID-19 was associated with a cognitive decline of over 12 months, whereas a mild form of the disease was only associated with this decline in the first six months [29]. However, the authors showed a higher risk of an early-onset cognitive decline independently of COVID-19 severity [29]. Accordingly, our results indicated a steeper continuous linear reduction in the MMSE score in those who had COVID-19, compared with controls, and a faster growth of the hazard of cognitive decline soon after COVID-19 diagnosis. Unfortunately, our results were limited within the first six months after COVID-19 diagnosis and did not consider some longer functional variations. Therefore, the yearly velocity of cognitive decline that we observed was a statistical extrapolation, and it should be taken with caution. Hypothetically, the speed of cognitive decline after this initial period could become equal to that of the controls, and the differences between the two groups could no longer exist. 

Based on previous studies, it would be expected that patients with severe COVID-19 may improve their cognitive functions in the long term [30,31]. This is an important point that implies that cognitive function should be assessed longer in patients with COVID-19 because of the chance of improvement. This should be considered with particular attention to patients with dementia, in whom, as we observed in our cohort, a worse cognitive function can lead to an increased incidence of new institutionalization. We would suggest that the lack of a long-term follow-up of cognitive function in elderly patients with dementia after COVID-19 recovery could miss some of such expected improvement and leave the elderly inappropriately assisted in long-term care institutes. Intensive rehabilitation after COVID-19 recovery could be a good strategy to prevent the potential long-term consequences that might permanently impact the life quality of elderly patients with cognitive impairment [32,33]. Further prospective studies should clarify this point. 

In this study, the MMSE score and ADL indexes were lowered on average in the cohort of patients, independently of COVID-19 occurrence. This observation was expected since all of our patients were subjected to the same social restriction imposed by the National Authority, and social restriction has been shown to impair the performance of the elderly, independently of COVID-19 [34,35,36]. Confinement and physical distancing as the consequence of social restrictions led to isolation and loneliness in aged people, which increased cognitive decline and reduced the ADL indexes in these subjects [5]. In addition, it has been shown that the confinement of patients with dementia increases their neuropsychiatric symptoms, such as delusion, agitation, irritability, appetite disturbance, and sleep disorders, which are associated with an increased cognitive decline and ADL impairment [35]. Additionally, home confinement affects the caregiver’s role by increasing the caregiving burden to assist patients with dementia and reducing the caregiver’s well-being status. All these factors may contribute to reducing the performance of patients with cognitive impairment, independently of COVID-19 occurrence [34,37]. 

Although in our study, the MMSE score declined independently of COVID-19, the MMSE score was lowered about twice faster in patients who had COVID-19. To justify this observation, it should be considered that SARS-CoV-2 infection has been associated with reduced brain size and lower gray matter thickness, blood–brain barrier damage, the inflammation of the brain tissue, and increased incidence of cerebrovascular ischemic and hemorrhagic events, encephalitis, and other metabolic brain abnormalities. All these events contribute to cognitive dysfunction and the development of dementia [12,38,39]. In addition, hospitalization for any cause is a known risk factor for cognitive dysfunction, new-onset dementia, ADL impairment, and reduced life quality [9,10]. For example, the hospitalization of the elderly with an acute illness increases the probability of complications that have been associated with cognitive decline and the future institutionalization of survivors, such as prolonged immobilization, reduction in plasma volume, polypharmacy, and delirium [40,41,42]. Although we did not have information about complications in our hospitalized patients, we considered hospitalization by itself a risk factor for cognitive impairment. Unexpectedly, in our study, hospitalization occurred only in COVID-19 patients; therefore, we could not differentiate the role of viral infection from that of hospitalization or its complications, and this represented a limitation of our study. We hypothesized that the lack of hospitalization in patients who did not contract COVID-19 was consistent with the phenomenon of reduced hospital admission, unless for COVID-19 reasons, during the first waves [43].

Other important limitations of this study were the following factors: First, the sample number of this study was small. During the pandemic, access to the memory care facility was limited because of home constraints and healthcare resources shifting to COVID-19 care. Therefore, fewer patients received memory care visits, and data availability was limited. Second, some baseline visits were performed years before the pandemic, with the risk of bias because of the different treatment periods. However, the level of assistance in the memory care facility is standardized, and it is independent of the period when the visit is performed. In addition, since all follow-up visits were performed in the pandemic period and the median time between the visits was similar in patients who had or did not have a significant cognitive decline, it means that for the outcome of this study, the different periods of baseline visits were not significant. Third, the retrospective inclusion of consecutive patients determined two groups of patients who had or did not have COVID-19 that were not homogenous. For example, the patients who had COVID-19 showed a trend of lower cognitive and ADL functioning and had higher medication use at the baseline. Therefore, as expected, those who contracted the infection already appeared as a frailer group [44]. We mitigated the impact of nonhomogeneity between the groups by weighting the effect of COVID-19 on the cognitive decline with the propensity score approach and by using multivariate analysis. Consistently, after correction, the effect of COVID-19 on cognitive decline and the MMSE score reduction remained strong. However, we could not account for those patients who had respiratory symptoms but were not tested for SARS-CoV-2 infection because we did not collect such information. This remained a potential source of bias, though we could consider the proportion of these patients to be very low because of the sensitivity of the population to the COVID-19 problem and its consequences, especially during the first waves. Fourth, we noticed a lower follow-up time in those patients who had COVID-19 than those who did not have the disease. This could have biased our results, since patients with COVID-19 may have visited earlier because of worsening cognitive function. Because we performed follow-up visits with a median time of 3.4 months after COVID-19 occurrence, we might have mainly observed the acute effect of the disease on cognitive function, losing important information about the long-term trajectory of cognitive change in these patients.

In conclusion, this study showed that COVID-19 was an important determinant of cognitive decline and accelerated cognitive dysfunction in elderly patients with pre-existing dementia followed up in an outpatient memory care facility. The clinical relevance of this observation was highlighted by the increased incidence of new institutionalization in patients with cognitive decline. Intensive rehabilitation and strict follow-up at memory care facilities might mitigate such a cognitive decline, but further long-term prospective studies should clarify this point. 

## Figures and Tables

**Figure 1 jcm-12-01845-f001:**
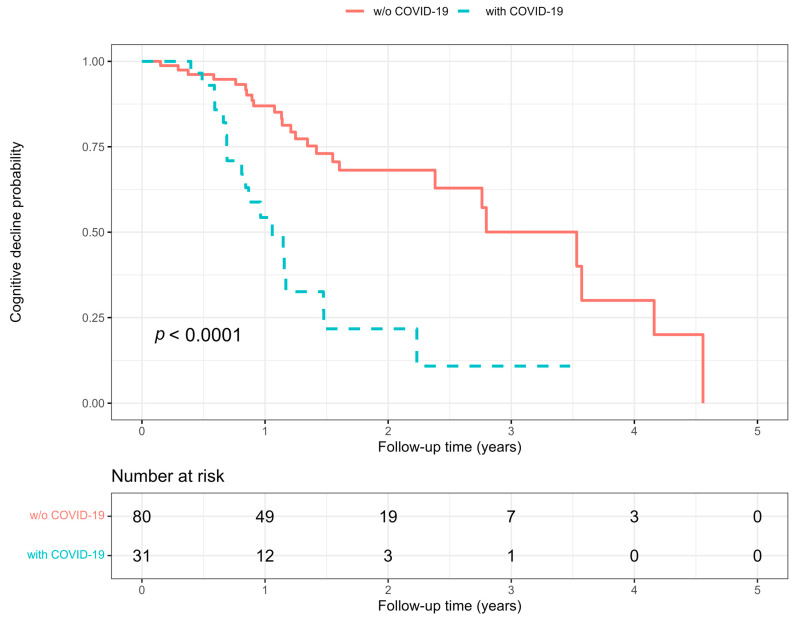
Kaplan–Meier curves of the probability of cognitive decline over time in elderly patients who had (cyan dashed line) or did not have (pink continuous line) COVID-19 between baseline and follow-up visits. Probability *p* was calculated with the nonparametric log-rank test. The number at risk are patients free from cognitive decline at a specific time point in either group.

**Figure 2 jcm-12-01845-f002:**
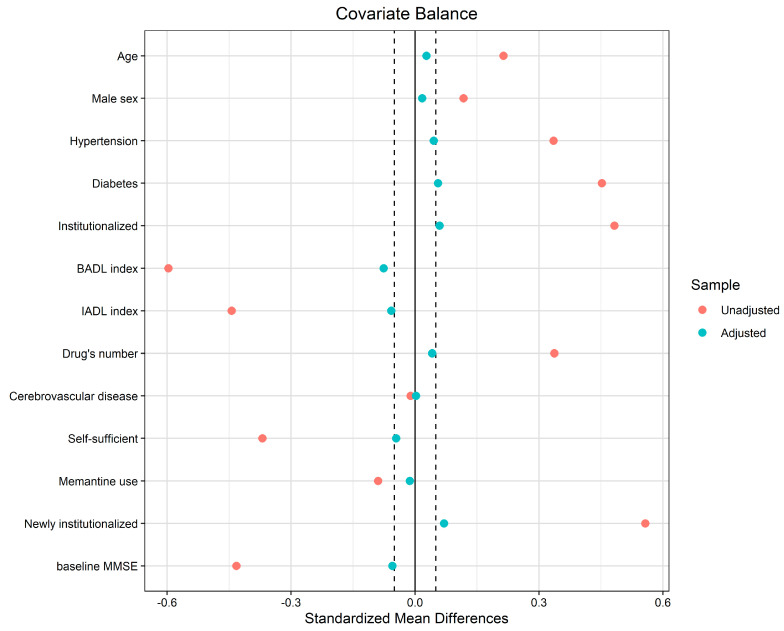
Graph of covariate balance before (unadjusted pink points) and after (adjusted cyan points) weighting variables using the propensity score model. For continuous variables, the standardized mean difference is reported, whereas, for binary variables, the raw mean difference is reported. Dark vertical dashed lines were plotted at ±0.05 from zero difference as a measure of significance.

**Figure 3 jcm-12-01845-f003:**
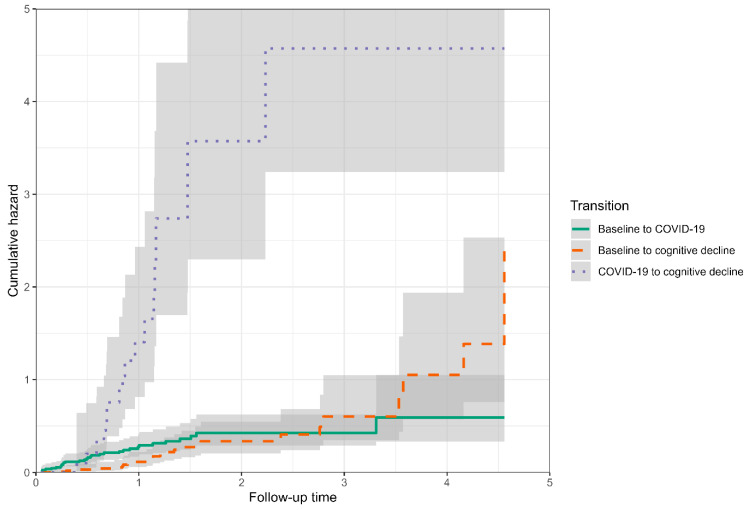
Graph of cumulative hazards of developing a cognitive decline over time in a 3-state model: from baseline to COVID-19 (green continuous line), from baseline to cognitive decline (pink dashed line), and from COVID-19 to cognitive decline (purple dotted line). Gray bands represent the 95% confidence interval.

**Figure 4 jcm-12-01845-f004:**
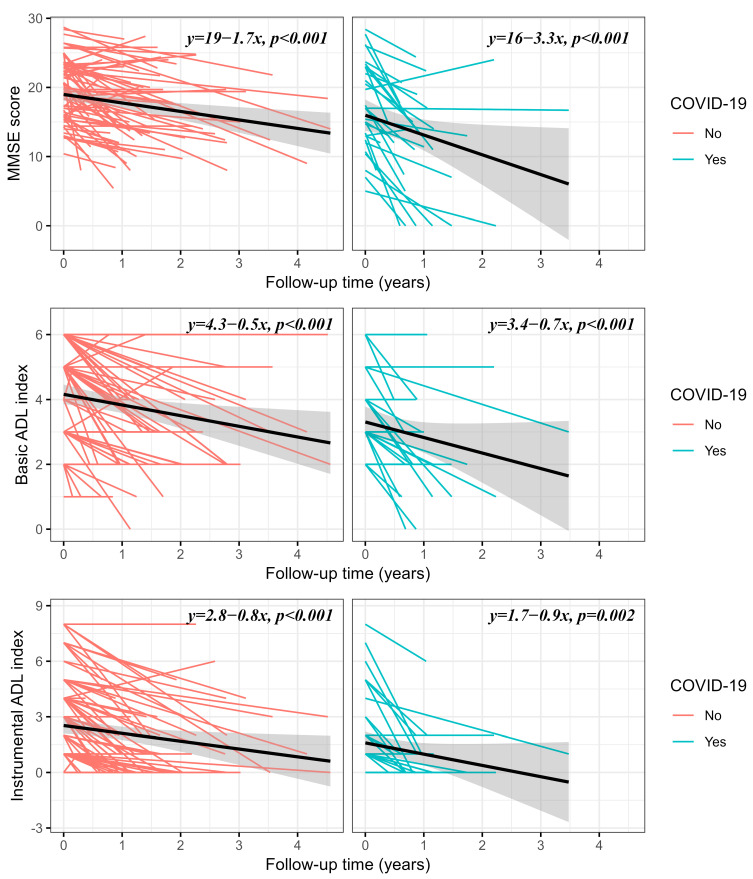
Changes between baseline and follow-up visits of MMSE score, BADL index, and IADL index over time with the summary regression line (black continuous line) and 95% confidence interval (gray bands) of patients who had (cyan lines) or did not have (pink lines) COVID-19. In the top right of each rectangle, the equation of the linear regression line is reported with the *p*-statistic of the slope.

**Table 1 jcm-12-01845-t001:** General clinical and laboratory characteristics of all patients and according to COVID-19 occurrence.

	All	Without COVID-19	With COVID-19	*p*
Patients (*n*)	111	80	31	-
Baseline variables				
Age (years)	82 ± 5	82 ± 5	83 ± 5	0.315
Male sex (*n* (%))	35 (32)	24 (30)	11 (36)	0.651
Hypertension (*n* (%))	66 (60)	44 (55)	22 (71)	0.138
Diabetes (*n* (%))	27 (24)	15 (19)	12 (39)	0.047
Dyslipidemia (*n* (%))	41 (37)	30 (38)	11 (36)	1.000
Cerebrovascular disease (*n* (%))	11 (9.9)	8 (10)	3 (9.7)	1.000
Cardiovascular disease (*n* (%))	37 (33)	26 (33)	11 (36)	0.824
Chronic kidney disease (*n* (%))	10 (9.0)	7 (8.8)	3 (9.7)	1.000
Parkinson’s disease (*n* (%))	7 (6.3)	7 (8.8)	0	0.187
Self-sufficient (*n* (%))	26 (23)	22 (28)	4 (13)	0.136
Institutionalized (*n* (%))	7 (6.3)	2 (2.5)	5 (16)	0.018
Dementia type (*n* (%)):				0.729
AD	28 (25)	21 (26)	7 (23)	
Vascular	25 (23)	19 (24)	6 (19)	
Mixed	37 (33)	24 (30)	13 (42)	
Other	21 (19)	16 (20)	5 (16)	
BADL index	5.0 [3.0, 5.5]	5.0 [3, 6]	3.0 [2, 4]	0.006
IADL index	2.0 [1.0, 4.0]	3.0 [1, 4]	1.0 [0, 3]	0.020
MMSE score	19.1 ± 5.0	19.8 ± 4.3	17.5 ± 6.3	0.027
Total drugs number	4.0 [2.0, 6.0]	3.5 [2.0, 6.0]	5.0 [3.0, 6.0]	0.107
Memantine (*n* (%))	5 (4.5)	4 (5.0)	1 (3.2)	1.000
Anticholinergic drug (*n* (%))	17 (15)	11 (14)	6 (19)	0.558
Antipsychotic drug (*n* (%))	22 (20)	15 (19)	7 (23)	0.791
Antidepressant drug (*n* (%))	23 (21)	16 (20)	7 (23)	0.797
Benzodiazepines (*n* (%))	21 (19)	16 (20)	5 (16)	0.790
Follow-up variables				
Follow-up time (years)	1.1 [0.7–1.7]	1.2 [0.8, 1.9]	0.9 [0.6, 1.2]	0.013
Change in MMSE score	−3.0 [−6.8, −1.4]	−2.7 [−5.8, −1.0]	−6.0 [−10, −3.2]	0.002
Significant cognitive decline (*n* (%))	44 (40)	25 (31)	19 (61)	0.005
Change in BADL index	−1.0 [−2.0, 0.0]	−1.0 [−2.0, 0.0]	−1.0 [−2.0, 0.0]	0.848
Change in IADL index	−1.0 [−2.5, 0.0]	−1.0 [−3.0, 0.0]	−1.0 [−2.0, 0.0]	0.235
Hospitalization for any cause (*n* (%))	6 (5.4)	0	6 (19)	<0.001
New institutionalization (*n* (%))	30 (27)	16 (20)	14 (45)	0.016

AD, Alzheimer’s disease; BADL, basal activity of daily living; IADL, instrumental activity of daily living, MMSE, Mini-Mental State Examination; *p*, probability.

**Table 2 jcm-12-01845-t002:** General clinical and laboratory characteristics of patients according to significant cognitive decline.

	Without Cognitive Decline	With Cognitive Decline	*p*
Patients (*n*)	67	44	-
Baseline variables			
Age (years)	82 ± 5	82 ± 6	0.734
Male sex (*n* (%))	19 (28)	16 (36)	0.409
Hypertension (*n* (%))	38 (57)	28 (64)	0.555
Diabetes (*n* (%))	14 (21)	13 (30)	0.367
Dyslipidemia (*n* (%))	26 (39)	15 (34)	0.690
Cerebrovascular disease (*n* (%))	4 (6.0)	7 (16)	0.109
Cardiovascular disease (*n* (%))	23 (34)	14 (32)	0.839
Chronic kidney disease (*n* (%))	5 (7.5)	5 (11)	0.514
Parkinson’s disease (*n* (%))	4 (6.0)	3 (6.8)	1.000
Self-sufficient (*n* (%))	21 (31)	5 (11)	0.021
Institutionalized (*n* (%))	3 (4.5)	4 (9.1)	0.432
Dementia type (*n* (%))			0.808
AD	18 (27)	10 (23)	
Vascular	13 (19)	12 (28)	
Mixed	23 (34)	14 (32)	
Other	13 (19)	8 (18)	
MMSE score	18.8 ± 4.4	19.6 ± 5.7	0.390
BADL index	5.0 [3.0, 5.5]	4.5 [3.0, 5.25]	0.963
IADL index	2.0 [1.0, 5.0]	2.0 [1.0, 4.0]	0.552
Total drugs number (*n*)	4.0 [2.0, 6.0]	4.0 [2.0, 6.0]	0.340
Memantine (*n* (%))	1 (1.5)	4 (9.1)	0.079
Anticholinergic drug (*n* (%))	10 (15)	7 (16)	1.000
Antipsychotic drug (*n* (%))	14 (21)	8 (18)	0.811
Antidepressant drug (*n* (%))	14 (21)	9 (21)	1.000
Benzodiazepine drug (*n* (%))	14 (21)	7 (16)	0.623
Follow-up variables			
COVID-19 (*n* (%))	12 (18)	19 (43)	0.005
Follow-up time (years)	1.1 [0.8, 1.8]	1.1 [0.7, 1.5]	0.602
Change in MMSE score	−2.0 [−3.0, 0.0]	−8.2 [−10.1, −6.0]	<0.001
Change in BADL index	0.0 [−1.0, 0.0]	−1.0 [−2.25, 0.0]	0.001
Change in IADL index	−1.0 [−2.0, 0.0]	−1.0 [−3.0, −1.0]	0.192
Hospitalization for any cause (*n* (%))	1 (1.5)	5 (11)	0.035
New institutionalization (*n* (%))	14 (21)	16 (36)	0.084

AD, Alzheimer’s disease; BADL, basal activity of daily living; IADL, instrumental activity of daily living, MMSE, Mini-Mental State Examination; *p*, probability.

**Table 3 jcm-12-01845-t003:** Predictors of COVID-19 and significant cognitive decline in univariate Cox proportional hazards analysis.

	COVID-19		Cognitive Decline	
Variable	HR (95% CI)	*p*	HR (95% CI)	*p*
Age (every 10 years)	1.43 (0.73, 2.80)	0.293	0.98 (0.56–1.71)	0.936
Male sex (yes/no)	1.39 (0.66, 2.91)	0.382	1.62 (0.86, 3.02)	0.133
Hypertension (yes/no)	2.12 (0.97, 4.63)	0.059	1.77 (0.94, 3.33)	0.078
Diabetes (yes/no)	2.31 (1.11, 4.78)	0.025	1.38 (0.71, 2.67)	0.344
Dyslipidemia (yes/no)	0.99 (0.47, 2.09)	0.988	1.07 (0.56, 2.03)	0.841
Cerebrovascular disease (yes/no)	1.00 (0.30, 3.30)	0.994	2.28 (0.99, 5.25)	0.053
Cardiovascular disease (yes/no)	1.29 (0.62, 2.69)	0.503	1.17 (0.61, 2.23)	0.635
Chronic kidney disease (yes/no)	1.06 (032, 3.52)	0.919	1.62 (0.63, 4.17)	0.319
Parkinson’s disease (yes/no)	-	-	0.98 (0.30, 3.22)	0.973
Self-sufficient (yes/no)	0.46 (0.16, 1.32)	0.148	0.42 (0.17, 1.08)	0.073
Institutionalized (yes/no)	2.86 (1.10, 7.48)	0.032	1.69 (0.60, 4.79)	0.320
Dementia type (yes/no):				
AD (Ref.)	1.00	-	1.00	-
Vascular	0.99 (0.33, 2.93)	0.980	1.52 (0.65, 3.54)	0.331
Mixed	1.58 (0.63, 3.97)	0.329	1.33 (0.58, 3.02)	0.498
Other	0.84 (0.26, 2.64)	0.759	0.68 (0.26, 1.80)	0.434
BADL (every 1 point)	0.66 (0.51, 0.85)	0.001	0.79 (0.64, 0.99)	0.039
IADL (every 1 point)	0.80 (0.67, 0.96)	0.014	0.86 (0.75, 0.98)	0.026
MMSE score (every 1 point)	0.92 (0.86, 0.99)	0.024	0.98 (0.92, 1.04)	0.548
New institutionalization (yes/no)	-	-	1.71 (0.91, 3.20)	0.096
Drug numbers (every 1 drug)	1.15 (1.00, 1.32)	0.054	1.13 (1.01, 1.27)	0.038
Memantine (yes/no)	0.83 (0.11, 6.12)	0.853	4.13 (1.42, 12.0)	0.009
Anticholinergic drug (yes/no)	1.63 (0.66, 4.01)	0.290	1.83 (0.79, 4.21)	0.157
Antipsychotic drug (yes/no)	1.43 (0.61, 3.34)	0.412	1.48 (0.67, 3.23)	0.330
Antidepressant drug (yes/no)	1.25 (0.54, 2.92)	0.604	1.30 (0.62, 2.76)	0.487
Anxiolytic drug (yes/no)	0.78 (0.30, 2.04)	0.616	0.67 (0.28, 1.60)	0.368
COVID-19 (yes/no)	-	-	3.94 (2.09, 7.43)	<0.001
Hospitalization for any cause (yes/no)	-	-	4.10 (1.59, 10.6)	0.003

AD, Alzheimer’s disease; BADL, basal activity of daily living; IADL, instrumental activity of daily living, MMSE, Mini-Mental State Examination; HR, hazards ratio; CI, confidence interval; *p*, probability.

**Table 4 jcm-12-01845-t004:** Effect of COVID-19, follow-up time, and their interaction (COVID x follow-up time) on MMSE, basic ADL, and instrumental ADL changes by linear mixed effect model.

	Dependent Variable		
	MMSE Score	Basic ADL	Instrumental ADL
Independent Variable	Estimate (95% CI)	Estimate (95% CI)	Estimate (95% CI)
A			
COVID-19 (yes/no)	−3.1 (−5.4, −0.9) **	−0.9 (−1.5, −0.2) **	−1.0 (−1.9, −0.2) *
Follow-up (year)	−1.7 (−2.3, −1.1) ***	−0.5 (−0.7, −0.4) ***	−0.8 (−1.0, −0.6) ***
COVID x Follow-up	−1.6 (−3.0, −0.2) *	−0.2 (−0.6, 0.2)	−0.1 (−0.6, 0.4)
B			
COVID-19 (yes/no)	−3.4 (−5.7, −1.0) **	−0.8 (−1.5, −0.1) *	−1.0 (−1.9, −0.05) *
Follow-up (year)	−1.7 (−2.2, −1.1) ***	−0.5 (−0.7, −0.4) ***	−0.8 (−1.0, −0.6) ***
COVID x Follow-up	−1.5 (−2.9, −0.06) *	−0.2 (−0.6, 0.2)	−0.1 (−0.6, 0.4)
C			
COVID-19 (yes/no)	−1.6 (−3.9, 0.7)	−0.6 (−1.3, 0.1)	−0.7 (−1.7, 0.3)
Follow-up (year)	−1.7 (−2.2, −1.1) ***	−0.5 (−0.7, −0.4) ***	−0.8 (−1.0, −0.6) ***
COVID x Follow-up	−1.5 (−2.9, −0.08) *	−0.2 (−0.6, 0.2)	−0.1 (−0.6, 0.4)

* *p* < 0.050; ** *p* < 0.010; *** *p* < 0.001. ADL, activities of daily living; CI, confidence interval. A. Unadjusted estimates. B. Estimates adjusted for sex, baseline age, baseline diabetes mellitus, baseline institutionalization, and new institutionalization. C. Estimates adjusted as in B plus hospitalization for any cause.

## Data Availability

Data are available from the corresponding author upon request and institutional permission of the University of Udine and the Department of Continuity of Care, ASUGI, Trieste.

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
