# Peer review of "COVID-19 Accelerated Cognitive Decline in Elderly Patients with Pre-Existing Dementia Followed up in an Outpatient Memory Care Facility"

_jcm, 2023, doi:10.3390/jcm12051845_

Round 1

Reviewer 1 Report

This is an interesting study following adults with dementia seen at a community memory clinic. The participants are split according to whether or not they were hospitalised due to Covid 19 during the follow up period. The analysis shows that those who had Covid 19 had greater cognitive decline during the follow up period than those who did not contract Covid 19 requiring hospitalisation. 

There are a number of points which need to be clarified. 

1) Did any participants contract Covid 19 & not become hospitalised? Was asymptomatic testing done? 

2) Can the authors tell if any of the participants who did baseline during the pandemic had Covid BEFORE they joined the study? This could significantly impact on the findings. 

3) The possibility of Covid 19 hospitalisation involving delirium is not discussed. It is well recognised that delirium of any cause is a major risk factor for cognitive decline & future institutionalisation. This point needs to be at least discussed in detail, and ideally the authors should provide data on whether participants experienced delirium. 

4) Did participants diagnosed before or during the pandemic receive the same level of memory clinic care? My experience is that access to diagnosis, treatment and social care was not the same. 

5) Why is a drop of 5 points on the MMSE chosen? 

6) I note that those who had Covid showed a trend to lower cognitive & ADL functioning and higher medication use at baseline. Therefore it is possible that those who caught Covid & were severely ill did so because they were already a frailer group. This should be discussed in more depth. 

Minor points - did all participants have capacity to give informed consent, or did relatives consent on their behalf? This should be clearer in the methods. 

'Anamestic' is an unusual term and confusingly often refers to an immunological response. I would avoid this term. 

There are a few confusing sentences - 3.2 'higher lowering' needs to be rephrased. 

Discussion - 'rarely the amelioration of the MMSE score reaches such an improvement' - I don't understand this. The sentence does not make sense in English and I don't know if the authors are mean a greater or lesser degree of improvement. 

Reviewer 2 Report

The researchers of this study found that cognitive decline was significantly higher among individuals with dementia who had COVID-19 when compared to those without COVID-19.

The findings are noteworthy and contribute to the literature. However, more details about the methods are needed.

The authors could do more to explain the specific contribution of their paper to the literature in the introduction section. How does it differ from the studies that the Lucia Crivelli et al. paper reviewed?

How were people in treated in the study if they had symptoms of COVID but did not test? Potentially, patients with the only the most severe COVID-19 were included in the study, as they were more likely to test. This potential source of bias could be mentioned in the discussion.

It appears to me that in Figure 2 that age, sex, and dementia severity are not used in the matching. I think these are important to include in the matching and corresponding diagram. If these are not included in the matching, I believe that analysis should be redone with these covariates included.

More information is needed in the data section regarding the cohort. It is a little unclear to me why patients with COVID were followed during a different time period than patients without COVID. Also, how many follow-ups on average did the patients receive?

The discussion section should mention the small sample size as a limitation.

Round 2

Reviewer 2 Report

I appreciate the careful consideration of my comments. I only have a couple of small modifications remaining that I would suggest:

1. I would describe everything you included in the propensity weighting in the methods section (in addition to the figure).

2. I would delete the following sentence: "A multicenter study could have increased the sample size, but it was not considered getting preliminary data"

I think it is sufficient to just say the sample size was limited by the data available.
